# SNA077, an Extract of Marine *Streptomyces* sp., Inhibits Melanogenesis by Downregulating Melanogenic Proteins via Inactivation of cAMP/PKA/CREB Signaling

**DOI:** 10.3390/ijms232314922

**Published:** 2022-11-29

**Authors:** Su-Jin Lim, Da-Won Jung, Prima F. Hillman, Sang-Jip Nam, Chang-Seok Lee

**Affiliations:** 1Department of Beauty and Cosmetic Science, Eulji University, Seongnam 13135, Republic of Korea; 2Department of Chemistry and Nanoscience, Ewha Womans University, Seoul 03760, Republic of Korea

**Keywords:** marine bacteria, *Streptomyces* sp., melanogenesis, melanogenic genes

## Abstract

Excess melanin in skin is known to be the main cause of hyper-pigmentary skin diseases such as freckles and lentigo. This study aimed to evaluate the depigmenting efficacy of an extract from the marine microorganism strain, *Streptomyces* sp. SNA077. To determine the anti-melanogenic efficacy of SNA077, we assessed the melanin contents of SNA077-treated B16, Melan-a, and MNT-1 cells. We observed the expression of key enzymes in melanogenesis via qRT-PCR and Western blot analyses. We further estimated the skin-whitening effect of SNA077 using a skin-equivalent model. SNA077 dramatically decreased the melanin production of B16 cells, Melan-a, and MNT-1 cells. In B16 cells treated with SNA077, the activity of cellular tyrosinase was clearly inhibited. In addition, the mRNA and protein expression levels of melanogenic genes were suppressed by SNA077 treatment in B16 and MNT-1 cells. Upstream of tyrosinase, the expression levels of phospho-CREB, phospho-p38, PKA activity, cyclic AMP production, and MC1R gene expression were inhibited by SNA077. Finally, SNA077 clearly showed a skin-brightening effect with a reduced melanin content in the skin tissue model. Collectively, our results suggest for the first time that an extract of marine *Streptomyces* sp. SNA077 could be a novel anti-melanogenic material for skin whitening.

## 1. Introduction

To improve hyper-pigmentary disorders, such as freckles and lentigo, many researchers have performed the isolation of depigmenting agents from plants. The development of raw materials with strong biological action but few side effects is a challenge for researchers, because a biologically active substance will typically have corresponding toxicity. Thus, there have been numerous studies have examined bioactive materials isolated from natural plants. Natural substances, which typically have relatively low toxicity, have recently gained favor as novel agents in the cosmetics industry. However, there have been very few reports on bioactive materials from microorganisms, and bioactive compounds produced by marine microorganisms are particularly unexploited. Therefore, from among the many natural substances known to have biological activity, we focus on whole extracts and secondary metabolites produced by marine microorganisms, which are known to have many pharmacologically active molecules [1]. Here, we report new results revealing that a crude extract from the marine *Streptomyces* sp., SNA077, has anti-melanogenic effects and might offer potential as a new functional pharmaceutical ingredient.

Abnormal skin pigmentations, such as melasma and freckles, represent skin pigmentary disorders that are caused by sun exposure and stress conditions, including hormone changes. One of the main hormones known to cause melanogenesis is alpha-melanocyte stimulating hormone (α-MSH) which is released from keratinocytes under stress conditions or exposure to ultraviolet (UV) radiation [2,3,4,5]. The released α-MSH binds to the melanocortin 1 receptor (MC1R) and stimulates intracellular cAMP in melanocytes. The stimulated intracellular cAMP activates transcription factors, such as microphthalmia-associated transcription factor (MITF) and its transcription factor, cAMP response element-binding protein (CREB), via the cAMP-dependent phosphokinase A (PKA) and p38 MAPK signaling pathways [6,7,8]. These transcription factors, especially MITF, trigger melanogenesis by binding to the promoter regions of the genes encoding melanogenesis-related proteins, tyrosinase, Tyrp-1, and Tyrp-2. Tyrosinase is the main enzyme of melanin production; it acts to hydroxylate l-tyrosine to l-3,4-dihydroxyphenylalanine (L-DOPA), and subsequently oxidizes L-DOPA to DOPA quinone. Tyrp-2 catalyzes the tautomerization of dopachrome to produce 5,6-dihydroxyindole-2-carboxylic acid (DHICA) and also oxidizes DHICA to produce carboxylate indole-quinone [9,10,11,12]. As these melanogenic factors mainly act to regulate melanogenesis, they have been targeted for the development of novel whitening agents [13].

In the present study, we measured melanin content to clearly show that SNA077 has anti-melanogenic effects in the human melanoma cell line, MNT-1, and the mouse melanocyte cell lines, Melan-a and B16. To define the relevant upstream molecular signaling pathway (s), we tested the ability of SNA077 to suppress the mRNA and/or protein expression levels of melanogenic factors, as well as its effects on the cAMP/PKA/CREB and p38 MAPK signaling pathways, which mediate melanogenesis. Finally, we checked the whitening efficacy of SNA077 using a three-dimensional (3D) skin-equivalent tissue model.

## 2. Results

### 2.1. Secondary Metabolite Profiling of SNA077 Extract

By comparing the UV and MS spectra of peaks from the SNA077 extract to those of an in-house spectral library (Figure 1), we identified the presence of several secondary metabolites, including the β-lactam antibiotic cephalosporin C, the cyclic decapeptide tyrocidine, the macrolide rapamycin, precingulid compounds, and members of the germicidin group. Two compounds with unique UV spectra (with retention times of 6.892 and 9.059 min) were also respectively observed. All the compounds indicated in the in-house UV spectral library were not isolated from the crude extract of SNA077. However, two compounds with the purity above 95% were isolated from the crude extract, and were identified as germicidin A (retention times of 8.939, *m*/*z* = 197.2 [M + H]^+^) and isogermicidin A (retention times of 9.333, *m*/*z* = 197.2 [M + H]^+^) by comparing the NMR spectral with the literature (Appendix A) [14,15]. *Streptomyces* sp. strain SNA077 was also expected to produce a red-colored pigment, given the red color of the extract. To identify the pigments within the SNA077 extract, fractionation was conducted. Seven fractions were obtained, and the seventh-fraction peak, with a retention time of 12.543 min, was identified as a member of the prodigiosin alkaloid family of natural products based on comparison to an in-house spectral library (match score = 998, Appendix A) [16] along with the comparison of its ^1^H NMR spectral with the literature (Appendix A). Prodigiosin, which is characterized by a deep, blood-red color [16], is reportedly produced by *Serratia marcescens*, *Pseudomonas magneslorubra*, *Alteromonas rubra*, and Gram-positive actinomycetes, such as *Streptomyces longisporus ruber* and *Streptoverticillium rubrireticuli*.

### 2.2. Inhibitory Effect of SNA077 on Melanin Content and Cellular Tyrosinase Activity in B16, Melan-a, and MNT-1 Cells

Prior to assessing the whitening efficacy of SNA077, we tested its cytotoxicity in melanocytes. In B16 cells, SNA077 at doses between 6.25 and 12.5 μg/mL did not show cytotoxicity, and only mild cytotoxicity was observed following treatment with 25 μg/mL (Figure 2A). Based on these results, non-cytotoxic doses of 2.5 to 10 μg/mL were subsequently used to determine the inhibitory effect of SNA077 on melanogenesis. In order to compensate for cellular amounts, melanin content was normalized with respect to the total protein content by performing a protein assay. Our results revealed that the application of 2.5 to 10 μg/mL SNA077 dose-dependently suppressed the extracellular and intracellular productions of melanin in α-MSH-stimulated B16 cells (Figure 2B,C). Kojic acid was used as a positive control for anti-melanogenic activity. To assess the effect of SNA077 in Melan-a melanocytes, cells were treated with SNA077 at doses ranging between 6.25 and 25 μg/mL. As seen for B16 cells, SNA077 did not affect cytotoxicity up to 12.5 μg/mL in Melan-a cells (Figure 2D), and the application of 2.5 to 10 μg/mL SNA077 dose-dependently inhibited the intracellular melanin content in Melan-a cells (Figure 2E). Similarly, we treated MNT-1 cells with SNA077 at doses ranging between 6.25 and 25 μg/mL. Unlike B16 cells, SNA077 showed cytotoxicity at 12.5 μg/mL in MNT-1 cells (Figure 2F); thus, we used SNA077 below 10 μg/mL concentration. The application of 2.5 to 10 μg/mL SNA077 dose-dependently inhibited the intracellular melanin content in MNT-1 cells (Figure 2G). Interestingly, SNA077 showed a stronger inhibitory effect on melanogenesis than kojic acid under the tested conditions. When viewed under a microscope, the SNA077-treated B16 and MNT-1 cells were brighter in color than the corresponding control groups (Figure 2H,I). To begin exploring the anti-melanogenic mechanism of SNA077, we measured direct tyrosinase activity using mushroom tyrosinase and also measured intracellular tyrosinase activity by using cellular tyrosinase extracted from B16 cells, depending on the degree of L-DOPA oxidation. Treatment with 2.5 to 10 μg/mL SNA077 had no inhibitory effect on mushroom tyrosinase activity, suggesting that SNA077 does not directly affect tyrosinase activity (Figure 2J). We then measured cellular tyrosinase activity using an α-MSH-treated B16 cell lysate that had been treated with SNA077 for 48 h. Interestingly, 10 μg/mL SNA077 inhibited cellular tyrosinase activity in α-MSH-treated B16 cells (Figure 2K). These results collectively revealed that SNA077 has an inhibitory effect on melanin synthesis in melanocytes and further suggested that the anti-melanogenic mechanism of SNA077 is derived from decreasing the gene expression level of tyrosinase (because it decreased the activity of cellular tyrosinase but not that of mushroom tyrosinase).

### 2.3. Inhibitory Effect of SNA077 on mRNA and Protein Expression Levels of Melanogenesis-Related Genes in B16 and MNT-1 Cells

To evaluate the hypothesis that SNA077 could affect the gene expression of tyrosinase, the expression levels of three main enzymes known to affect melanogenesis, namely tyrosinase, Typr-1, and Tyrp-2, were analyzed. SNA077 was applied to α-MSH-stimulated B16 or MNT-1 cells at 2.5 to 10 μg/mL for 24 h, and the mRNA expression levels of tyrosinase, Tyrp-1, and Tyrp-2 were detected with real-time PCR. The results showed that SNA077 dose-dependently suppressed the mRNA expression levels of tyrosinase, Tyrp-1, and Tyrp-2 in B16 cells (Figure 3A) and MNT-1 cells (Figure 3B). Next, the protein expression levels of these melanogenesis-related factors in α-MSH-stimulated B16 and MNT-1 cells treated with the indicated concentrations of SNA077 for 48 h were detected using Western blot analysis. The results revealed that the protein expression levels of tyrosinase, Tyrp-1, and Tyrp-2 were dose-dependently decreased in B16 cells (Figure 3C–F) and MNT-1 cells (Figure 3G,H).

To further investigate the regulatory mechanism governing gene expression levels of these melanogenic enzymes, we assessed the expression of their major transcription factor, MITF, using real-time PCR and Western blot analysis. The application of SNA077 at the indicated concentration decreased the mRNA expression level of MITF after 24 h in B16 and MNT-1 cells (Figure 4A,B) and decreased the protein expression level of MITF after 36 h in B16 cells (Figure 4C,D) and after 48 h in MNT-1 cells (Figure 4E,F). These results collectively revealed that SNA077 has inhibitory effects on the expression levels of three main melanogenic enzymes and their transcription factor, MITF, in melanocytes. We observed correlations between the levels of intracellular tyrosinase activity and melanogenic gene expression, indicating that SNA077 decreased the melanin content by down-regulating expression levels of tyrosinase genes, and thereby inhibiting intracellular tyrosinase activity.

### 2.4. Anti-Melanogenic Effect of SNA077 Involves Down-Regulating the Phosphorylation of cAMP/PKA/CREB and MAPK in α-MSH-Stimulated B16 Cells

To explore the upstream signaling pathway involved in the down-regulation of MITF and tyrosinase in SNA077-treated melanocytes, we focused on the PKA/CREB and MAPK signaling pathways. The protein expression levels of CREB and MAPK were examined by Western blot analysis. The results showed that the ratio of phosphorylated CREB (p-CREB)/to total CREB (t-CREB) was significantly and dose-dependently decreased in α-MSH-stimulated B16 cells treated with 2.5 to 10 μg/mL SNA077 for 24 h (Figure 5A,B). To assess PKA activity in SNA077-treated melanocytes, the protein expression level of phospho-PKA substrate was investigated. α-MSH-stimulated B16 cells were treated with SNA077 for 8 h, and the protein level of phospho-PKA substrate was determined by Western blot assay. The results showed that expression level of phospho-PKA substrate was dose-dependently decreased by SNA077 (Figure 5C), indicating that PKA is inactivated by SNA077 treatment in melanocytes. Since PKA activity mainly depends on cAMP [4], we next measured intracellular cAMP levels using a commercial assay kit. We found that intracellular cAMP levels were significantly and dose-dependently decreased by SNA077 in α-MSH-stimulated B16 cells (Figure 5D), indicating that the anti-melanogenic mechanism of SNA077 could reflect inactivation of cAMP/PKA/CREB signaling. To further assess this potential anti-melanogenic signaling, we used real-time qPCR to assess the expression of MC1R, which is both the receptor of α-MSH and an upstream component of cAMP/PKA signaling. As expected, SNA077 treatment of α-MSH-stimulated B16 cells dose-dependently repressed the mRNA expression of MC1R (Figure 5E). We also investigated the protein expression levels of the MAPKs (ERK, p38, and JNK) in α-MSH-stimulated B16 cells treated with 2.5 to 10 μg/mL SNA077 for 24 h. The results showed that the ratio of p-JNK/t-JNK was not significantly affected, whereas the p-ERK/t-ERK and p-p38/t-p38 ratios were dose-dependently decreased by SNA077 in α-MSH-stimulated B16 cells (Figure 5F,G).

### 2.5. Inhibitory Effect of SNA077 on Melanogenesis in a 3D Human Skin Equivalent

To assess whitening effects in an in vivo-like condition, 3D human skin models have been widely used. To estimate whether SNA077 could decrease the degree of pigmentation, MelanoDerm was used. The MelanoDerm was topically treated with 0.03% SNA077 for 14 days and the color alteration was photographed. We found that SNA077 brightened the color of MelanoDerm compared to that of the non-treated control MelanoDerm (Figure 6A). Moreover, Fontana Masson’s staining showed that SNA077 decreased the level of intracellular melanin in this system (Figure 6B). These results showed that SNA077 could potentially decrease epidermal pigmentation.

## 3. Discussion

Many researchers have sought to discover novel active compounds for the treatment of hyperpigmentation-associated disorders. Many candidates have been assessed as potential anti-melanogenic agents. However, although many active compounds have whitening effects, their use in humans has been limited by side effects, such as cytotoxicity. Recently, natural substances have emerged as promising anti-whitening candidates because they have relatively few side effects compared to synthetic chemicals. However, most natural plant extracts are difficult to use in practice because raw materials are obtained in relatively limited quantities. One potential key to solving these problems is the use of microorganisms. Although microorganisms also face limitations, their natural products can be mass produced under optimized culture and extraction conditions. To date, various microorganisms, ranging from plant-derived microbial extracts to soil-harvested Cyanobacteria have been reported for their whitening efficacy and developed for cosmetic applications [17,18,19]. Recently, marine substances have been broadly researched for potential cosmetic applications, because marine resources are abundant and offer many effective ingredients. For example, seaweeds have been shown to have anti-aging, antioxidant, and anti-melanogenic effects [20]. The present study was conducted with a special focus on marine microorganisms as sources of skin-whitening agents. Relatively few specific experimental and mechanistic studies have examined the anti-melanogenic efficacy of marine microorganism extracts. Therefore, we studied both the anti-melanogenic effect and underlying mechanism of an extract from the marine microorganism, SNA077.

Melanogenesis occurs to protect the skin against external factors, such as UV light. Melanocytes produce melanin via the oxidative reactions of melanogenic proteins, including tyrosinase, Tyrp-1, and Tyrp-2 [9,10]. The expression of these melanogenic proteins is regulated by MITF, whose expression is mainly influenced by the cAMP-PKA-CREB pathway under stimulation by α-MSH [4,5,6]. α-MSH acts as a ligand for MC1R and stimulates adenylyl cyclase, thereby activating the cAMP-PKA-CREB pathway for potent melanogenesis [21,22]. Here, we reported that SNA077 can decrease the intracellular cAMP level in melanocytes (Figure 3D), suppressing PKA and CREB activity and thereby decreasing melanogenesis via inactivation of cAMP-PKA-CREB signaling. Interestingly, we found that the MC1R mRNA level was decreased after only 5 min of SNA077 treatment. Although this decrease in MC1R expression had a direct effect on the cAMP level in our system, representing an experimental limitation of the present study, we propose that SNA077 is likely to suppress the cAMP level at least partly via inhibiting MC1R expression while melanin is synthesized in melanocytes. In addition, we suggest the possibility that SNA077 could inhibit adenylate cyclase or some intracellular signal needed to activate adenylate cyclase, although the additional experiments such as adenylate cyclase activity will be needed.

In addition to the cAMP-PKA-CREB signaling pathway, α-MSH-induced melanogenesis also depends on MAPK signaling [23,24]. Among the MAPK family members, it is well known that inhibition of p38 can decrease MITF [25]. In addition, phospho-ERK has two different impacts on MITF: ERK is reportedly related to both upregulation of MITF gene expression and MITF degradation [26,27,28,29,30]. In our experiments, the melanin content was not decreased by the ERK inhibitor, PD98059, and similar inhibitions of melanin content were seen in cells treated with SNA077 plus PD98059, and with SNA077 alone. These data indicate that SNA077 decreased melanogenesis independent of the ERK pathway.

In the previous report, the treatment of cells with alpha-MSH led to the time-dependent phosphorylation of both ERK and p38 MAP kinases. However, only inhibition of p38 MAP kinase activity blocked the alpha-MSH-induced melanogenesis, indicating that activation of the p38 pathway can potently promote melanogenesis [6]. In the current study, we observed that SNA077 strongly suppressed p38 phosphorylation in melanocytes (Figure 4F). Therefore, our findings suggest that SNA077 appears to inhibit melanogenesis via inactivation of the p38 signaling pathway, but not the ERK pathway. Finally, beyond demonstrating the anti-melanogenic effect of SNA077 in vitro, we also observed this effect in a human skin equivalent. The maximum non-cytotoxic concentration of SNA077 in vitro was 10 μg/mL (0.001%), whereas that used in the skin equivalent system was 0.03%. For cosmetic application as a whitening material, the safe and effective concentration of a material is generally higher than that tolerated in culture, due to issues with skin permeability and material stability. As shown in Figure 5, 0.03% SNA077 appeared to effectively whiten human artificial skin, suggesting that it could be effective in a clinical test for depigmentation.

Collectively, our present study provides preliminary evidence suggesting that SNA077 could be a potent and effective whitening agent for use in cosmetics and medicinal formulations. In future studies, we will seek to identify and characterize the effective compound (s) in SNA077 extracts.

## 4. Materials and Methods

### 4.1. General Experiments

Low-resolution LC/MS measurements were performed using an Agilent Technologies 1260 quadrupole (Agilent Technologies, Santa Clara, CA, USA) and a Waters micro mass ZQ LC/MS system (Waters Corp, Milford, MA, USA) with a reverse-phase column (Phenomenex Luna C18 (2) 100 Å, 50 mm × 4.6 mm, 5 µm, Torrance, CA, USA). The flow rate was 1.0 mL/min, the duration was 18.0 min, and the solvent gradient ranged from 5% to 100% acetonitrile (CH_3_CN, Samchun, Seoul, Republic of Korea) in water. LC/MS was performed at the National Research Facilities and Equipment Center (NanoBioEnergy Materials Center) at Ewha Womans University (Seodaemun, Seoul, Republic of Korea).

### 4.2. Collection, Phylogenetic Analysis, Cultivation, Extraction, and Fractionation

Strain SNA077 was isolated from Yeosu Manseongni Black Beach in South Korea, and identified as a *Streptomyces* sp. based on 16S rRNA gene sequence analysis (accession no. EU841563.1). Marine-derived *Streptomyces* sp. strain SNA077 was cultured in 2.8 L Fernbach flasks containing 1 L of seawater-based medium (10 g starch, 2 g yeast extract, 4 g peptone, and 34.74 g artificial sea salt dissolved in distilled water) and shaken at 27 °C at 120 rpm. After 7 days of cultivation, the culture broth was extracted using ethyl acetate (EtOAc) and the solvent was removed in vacuo to obtain SNA077 extract. The crude extract of SNA077 was fractionated by reverse-phase medium-pressure liquid chromatography on C-18 resin with a step gradient of water and methanol (MeOH; Samchun, Seoul, Republic of Korea), using a solvent gradient of 10% to 100% MeOH.

### 4.3. Secondary Metabolite Profiling Analysis

The crude extracts of strain SNA077 were analyzed by liquid chromatography (LC)-mass spectrometry (MS). LC traces were generated at 210, 254, 280, and 310 nm, and the UV absorbance spectra associated with each peak were evaluated by comparing the peaks to an in-house spectral library (Appendix A). Peaks were assigned to a particular compound class if they had a UV matching score of 900 or greater, as reported using the software provided with the Agilent Technologies ChemStation (Santa Clara, CA, USA). The obtained positive scores ranged from 903 to 999.

### 4.4. Cell Culture

The B16 mouse melanoma cell line was cultured in Dulbecco’s modified Eagle’s medium (DMEM; Welgene, Gyeongsan-si, Republic of Korea) supplemented with 5% fetal bovine serum (FBS; ATCC, Manassas, VA, USA) and 1% Penicillin-Streptomycin Mixture (PS; Lonza, Basel, Switzerland) and used at passages 5–30. The Melan-a murine melanocyte cell line was cultured in RPMI 1640 medium (RPMI 1640; Lonza, Walkersville, MD, USA) containing 10% fetal bovine serum (FBS; Gibco, Grand Island, NY, USA), 1% PS, and 200 nM phorbol 12-myristate 13-acetate (PMA; Sigma-Aldrich, St. Louis, MO, USA) and used at passages 1–10. MNT-1 cells, representing a highly pigmented human melanoma cell line, were cultured in minimum essential medium (MEM; Gibco) containing 10% DMEM, 20% FBS (FBS; Gibco, Grand Island, NY, USA), 1% PS, and 1% N-(2-hydroxyethyl)piperazine-N′-(2-ethanesulfonic acid) (HEPES, 1M; Gibco), and used at passages 5–30. All cells were cultured at 37 °C in a humidified environment with 5% CO_2_ and 95% air.

### 4.5. Cell Viability Assay

Cell viability was determined using Quanti-MAX™ WST-8 Cell Viability Assay Kit (Biomax, Seoul, Republic of Korea) according to the manufacturer’s protocols. Cells were seeded in a 96-well plate at an appropriate concentration for 24 h, and then the culture medium was replaced with fresh culture medium containing SNA077 diluted to the indicated concentrations. Following exposure for 72 h, cell viability was assessed by replacing the medium with the appropriate medium containing 10% WST-8 solution. The plate was incubated for 1 h and the absorbance was measured at 450 nm using a Microplate Reader (BioTek, Winooski, VT, USA). Cell viability was computed with the general formula: cell viability (%) = optical density (450 nm) of the experimental group/optical density of the control group × 100.

### 4.6. Measurement of Melanin Content

For melanin quantification, B16 cells (2 × 10^4^ cells/well in 48-well plates), Melan-a cells (10 × 10^4^ cells/well in 24-well plates), and MNT-1 cells (10 × 10^4^ cells/well in 24-well plates) were cultured for 24 h. B16 cells were treated with increasing concentrations of SNA077 in the presence of α-MSH (0.1 μM) to induce melanin production in phenol red-free cell culture medium because phenol red would interfere with observation of secreted extracellular melanin. After 72 h, the extracellular melanin content in the phenol red-free cell culture medium of B16 cells was measured at 405 nm using a Synergy™ HTX Multi-Mode Microplate Reader (Winooski, VT, USA). For measurement of intracellular melanin contents, 1N NaOH was added to each well, the plate was heated to 60 °C for 30 min, the resulting lysate was aliquoted to a 96-well plate, and absorbance was determined at 405 nm. The protein concentration of each lysate was determined with a Pierce BCA Protein Assay Kit (Thermo Fisher Scientific, Waltham, MA, USA). The intracellular melanin content was normalized with respect to the total protein amount. Melanin levels were calculated by comparison to those in the corresponding controls and are shown as percentages.

### 4.7. Mushroom Tyrosinase Activity Assay

Mushroom tyrosinase activity was estimated by measuring oxidase activity. The reaction mixture consisted of 5 μL of sodium phosphate buffer (0.1 M), pH 6.8, 40 μL of distilled water, and 5 μL of enzyme solution (mushroom tyrosinase, 2000 units in 0.1 M PBS; Sigma Aldrich, T3824). SNA077 was dispensed to a 96-well plate at 5 μL/well, and the prepared reaction mixture plus 50 μL of 20 mM L-DOPA or 0.03% L-tyrosine (tyrosinase substrate) was added. The level of dopachrome in the mixture was determined at 475 nm in the spectrophotometer at least 5 times over 10 at an incubation temperature of 37 °C.

### 4.8. Cellular Tyrosinase Activity Assay

B16 cells (20 × 10^4^ cells/well in 6-well plates) were seeded and incubated at 37 °C in a humidified environment with 5% CO_2_ for 24 h. Next, kojic acid (100 μg/mL, positive control) or SNA077 (10 μg/mL) were added and the plates were incubated for 48 h. After 48 h incubation, the cells were washed with cold PBS and lysed in 1 mL of sodium phosphate buffer (0.1 M, pH 6.8) containing 1% Triton X-100 for 20 min. The lysates were centrifuged at 13,000 RPM at 4 °C for 20 min. Except debris, the supernatant and the substrate solution (L-DOPA, 10 mM) in equal volume were dispensed to a 96-well plate. To determine intracellular tyrosinase activity in the supernatant (i.e., the reaction with L-DOPA), absorbance was measured at 490 nm every 10 min for at least 1 h at 37 °C, using a spectrophotometer.

### 4.9. Western Blot Analysis

B16 or MNT-1 cells were cultured in 6-well plates and treated with SNA077 as described in the corresponding figure legend. The following day, cells were washed twice with cold PBS and total intracellular proteins were extracted in RIPA buffer 1X (Cell Signaling Technology, Danvers, MA, USA) supplemented with a 1:200 dilution of protease inhibitor cocktail III and phosphatase inhibitor cocktail set III (Calbiochem Biosciences, La Jolla, CA, USA) for 20 min on ice. After centrifugation at 13,000 RPM at 4 °C, the supernatant was collected. A Pierce BCA Protein Assay Kit was used to estimate the protein quantity of extracts. Equal amounts of soluble proteins were resolved by 10% SDS-PAGE and transferred to a nitrocellulose membrane (BioRad, Hercules, CA, USA) in cold transfer buffer (25 mM Tris, 192 mM glycine, and 20% (*w*/*v*) MeOH) for 90 min at 280 mA. Membranes were blocked for at least 1 h with TBS 1X supplemented with 5% Blotting-Grade Blocker (BioRad) and incubated overnight at 4 °C with primary antibodies diluted to the proper concentration (per the provided data sheet) in 1X TBS. Following an overnight incubation, the membranes were further incubated with secondary antibodies and enhanced using a Clarity™ Western ECL substrate (ECL solution; BioRad). Images of the blotted membranes were captured with an iBright™ CL750 Imaging System (Invitrogen, Carlsbad, CA, USA). Equal loading was assessed by comparison to the endogenous actin protein. Densitometric analysis was performed using ImageJ program (National Institutes of Health, Bethesda, MD, USA).

### 4.10. Real-Time PCR

To determine the expression levels of mRNAs encoding melanogenesis-related factors, B16 or MNT-1 cells were cultured in 6-well plates and treated with SNA077 as described in the corresponding figure legends. The following day, cells were washed twice with cold PBS and total RNA was extracted from cell lysates using the TRIzol reagent (Thermo-Fisher Scientific) according to the manufacturer’s instructions. Complimentary DNA was synthesized from total mRNA using the GoScript™ Reverse Transcription System (Promega, Madison, WI, USA) and a thermal cycler (T-100; BioRad). The relative expression level of target cDNA was detected using the CFX Connect Optics Module (BioRad) and iQ SYBR^®^ Green Supermix (BioRad). All results were normalized with respect to the mRNA level of β-actin. The specific primers were purchased from BioRad.

### 4.11. Cyclic AMP Assay

The cAMP levels in cell lysates were estimated using a cAMP Parameter assay kit (R&D Systems, Minneapolis, MN, USA). B16 cells (20 × 10^4^ cells/well in 24-well plates) were seeded and allowed to incubate at 37 °C in a humidified environment with 5% CO_2_ for 24 h. The B16 cells were then pretreated with SNA077 (2.5–10 μg/mL) for 4 h, co-treated with SNA077 (2.5–10 μg/mL) and α-MSH (0.1 μM) for 10 min, and then washed three times in cold PBS. The cells were resuspended in Cell Lysis Buffer 5 in cAMP parameter assay kit (diluted 1:5), the freeze/thaw cycle was repeated, and the lysates were centrifuged at 13,000 RPM for 10 min. Supernatants were collected and directly used for the cAMP assay procedure according to the manufacturer’s protocol.

### 4.12. Skin Whitening Test Using a MelanoDerm Skin Equivalent

MelanoDerm (MEL-300-B; MatTek Corp., Ashland, MA, USA), which was used for the skin-lightening study, was cultured in EPI-100-NMM-113-PRF medium (MatTek Corp.) at 37 °C in a humidified environment with 5% CO_2_. SNA077 was topically applied to the MelanoDerm once every 3 days for 14 days. Thereafter, the degree of epidermal pigmentation was examined by optical estimates and histological analyses. Melanin was subjected to Fontana Masson’s (FM) staining and histologically examined.

### 4.13. Statistical Analysis

Data are expressed as the mean ± standard deviations (SDs), and statistical significance was analyzed by the Student’s *t*-test. *p*-values less than 0.05 were considered statistically significant.

## 5. Conclusions

Collectively, the data from this study provide preliminary evidence supporting the possibility that SNA077 may represent a potent and effective whitening agent that could be used in cosmetics and medicinal formulations.

## Figures and Tables

**Figure 1 ijms-23-14922-f001:**
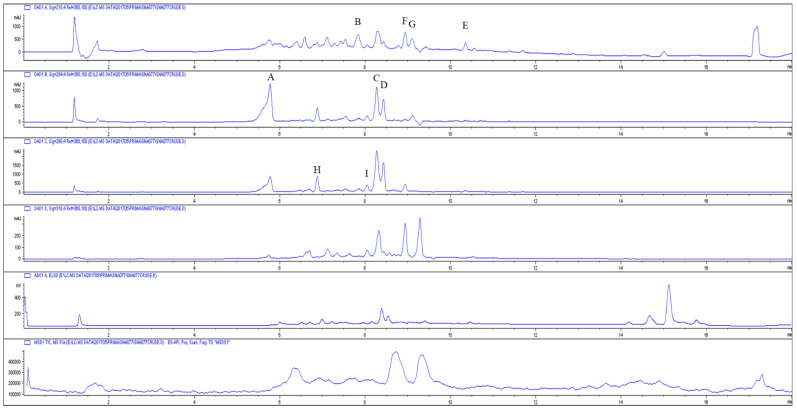
UV, ELSD, and mass spectra for compounds in culture extracts of SNA077 (blue) in comparison with known in-house compounds (red). (**A**) cephalosporin C, match score = 958; (**B**) tyrocidine, match score = 946; (**C**,**D**) Rapamycin, match score = 948, 961; (**E**) preechinulin, match score = 977; (**F**,**G**) germicidin A, match score = 971, 951; (**H**,**I**) staurosporin, match score 842, 870. Match scores higher than 900 were considered positive identification for that compound class.

**Figure 2 ijms-23-14922-f002:**
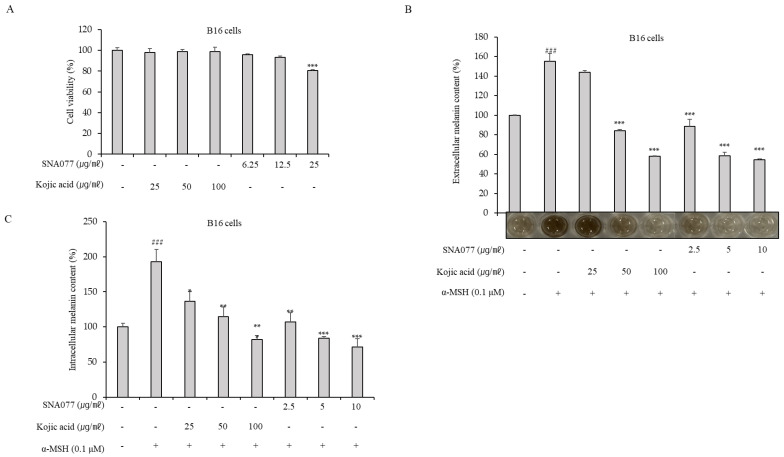
Anti-melanogenic efficacy of non-toxic concentrations of SNA077 in melanocytes. (**A**) Effect of SNA077 on B16 cell viability. B16 cells were treated with the indicated concentrations of SNA077 for 72 h, and cell viability was estimated using a CCK-8 assay kit. (**B**,**C**) Effects of a 72 h incubation with the indicated concentrations of SNA077 or kojic acid on the intracellular and extracellular melanin contents in α-MSH-stimulated B16 cells. (**B**) Extracellular melanin content was colorimetrically determined. Photographs show the color of the medium in each well following treatment with α-MSH and SNA077. (**C**) Intracellular melanin content was determined by measuring the absorbance of B16 cell lysates at 405 nm and normalizing the obtained values to the protein quantity, which was determined using a protein assay kit. Results are presented as means ± SD, expressed as a percentage relative to the control group. (**D**,**F**) Viability of Melan-a cells (**D**) and MNT-1 human melanoma cells (**F**) after treatment with SNA077 for 72 h. (**E**,**G**) Representative results show melanin contents in Melan-a cells (**E**) and MNT-1 cells (**G**) treated with SNA077 or kojic acid for 72 h. (**H**,**I**) Microscopy of B16 cells (**H**) and MNT-1 cells (**I**). (**J**,**K**) Mushroom **(J**) and cellular (**K**) tyrosinase activities were spectrophotometrically measured at 475 nm, as described in Materials and Methods. Figures depict results representative of at least three replicate experiments. Results are presented as means ± SD, expressed as a percentage relative to the control group (* *p* < 0.05, ** *p* < 0.01, *** *p* < 0.001 vs. control group). “-” means non-treated or without any treatment and “+” means treated with the indicated compound along with α-MSH at 0.1 μM where indicated. ### *p* < 0.001.

**Figure 3 ijms-23-14922-f003:**
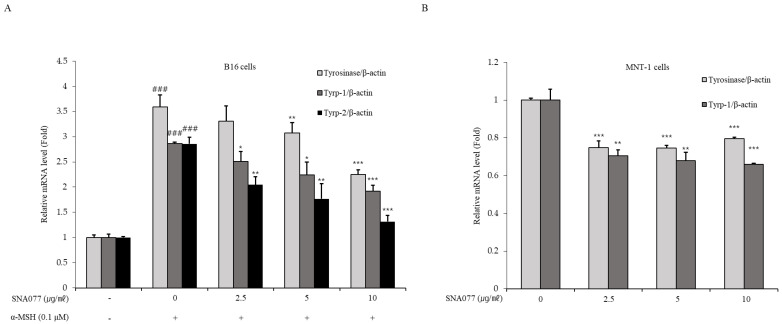
Inhibitory effect of SNA077 on the expression of melanogenic enzymes. (**A**,**B**) mRNA levels of melanogenic enzyme-encoding genes were determined by real-time qPCR analysis in B16 cells (**A**) and MNT-1 cells (**B**). Cells were treated with SNA077 for 24 h. The mRNA levels of the genes encoding tyrosinase, Tyrp-1, and Tyrp-2 were normalized to that of β-actin. (**C**,**E**,**G**) The protein levels of melanogenic enzymes were determined by Western blot analysis in B16 cells (**C**,**E**) and MNT-1 cells (**G**). Cells were treated with SNA077 for 72 h. Equal amounts of proteins were resolved by 10% SDS-PAGE and detected using specific antibodies. β-Actin was detected as a loading control. (**D**,**F**,**H**) Protein bands were densitometrically quantified and analyzed using ImageJ software. Results for B16 cells (**D**,**F**) and MNT-1 cells (**H**) are presented as means ± SD, expressed as a percentage relative to controls (* *p* < 0.05, ** *p* < 0.01, *** *p* < 0.001 vs. control group; ### *p* < 0.001 vs. the α-MSH-untreated control group).

**Figure 4 ijms-23-14922-f004:**
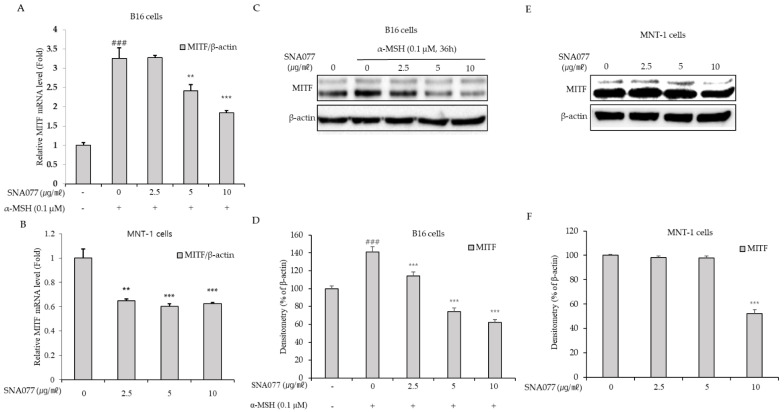
Inhibitory effect of SNA077 on the gene expression of MITF in melanocytes. (**A**,**B**) The mRNA level of MITF was determined by real-time qPCR in B16 cells (**A**) and MNT-1 cells (**B**). Cells were treated with SNA077 for 24 h. The mRNA level of MITF was normalized to that of β-actin. (**C**,**E**) The protein level of MITF was determined by Western blot analysis in B16 cells (**C**) and MNT-1 cells (**E**). Cells were treated with SNA077 for 36 h or 48 h. Equal amounts of proteins were resolved by 10% SDS-PAGE and detected using specific antibodies; β-actin was detected as a loading control. (**D**,**F**) Protein bands were densitometrically quantified and analyzed using ImageJ software. Results for B16 cells (**D**) and MNT-1 cells (**F**) are presented as means ± SD, expressed as a percentage relative to controls (** *p* < 0.01, *** *p* < 0.001 vs. control group; ### *p* < 0.001 vs. the α-MSH-untreated control group).

**Figure 5 ijms-23-14922-f005:**
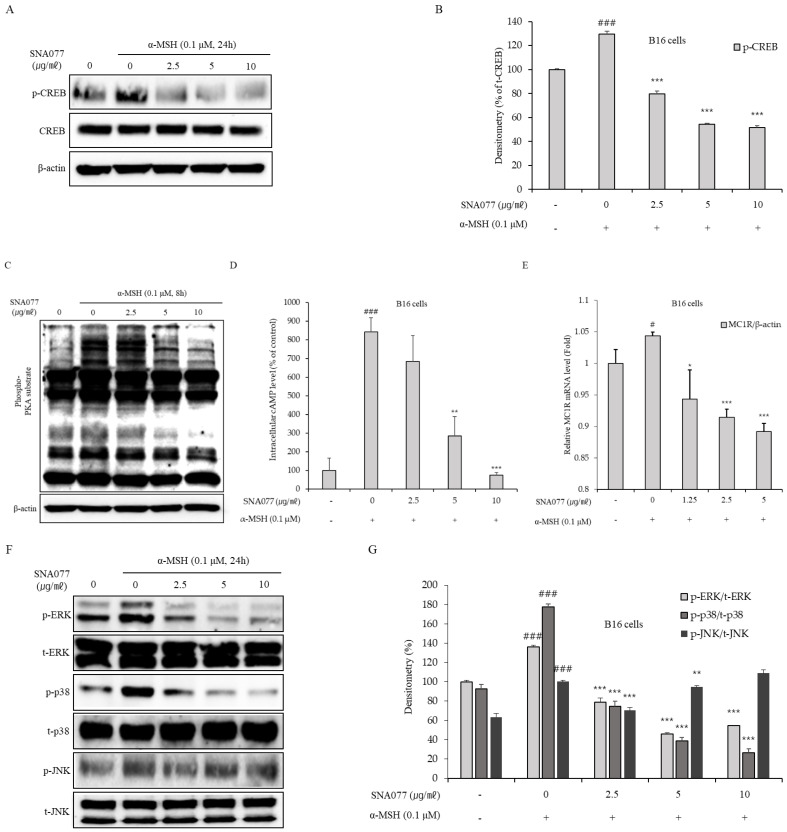
Inhibitory effect of SNA077 on α-MSH-induced cAMP/PKA/CREB and MAPK signaling in B16 cells. (**A**,**C**) Protein levels of phospho-CREB (**A**) and phospho-PKA substrate (**C**) were determined by Western blot analysis in B16 cells. Cells were treated with SNA077 for the indicated times. Equal amounts of proteins from B16 cells were resolved by 10% SDS-PAGE and detected using specific antibodies. β-Actin was detected as a loading control. (**B**) Protein bands were densitometrically quantified and analyzed using ImageJ software. To study upstream signaling, B16 cells were treated with SNA077 for 15 min and the intracellular cAMP levels in lysates were estimated using a cAMP Parameter assay kit (**D**). (**E**) Based on the cAMP test results, B16 cells were treated with SNA077 for 10 min and mRNA level of MC1R was determined by real-time qPCR analysis. (**F**) Protein levels of phospho-MAPKs were determined by Western blot analysis in B16 cells. Cells were treated with SNA077 for 24 h. Equal amounts of proteins were resolved by 10% SDS-PAGE and detected using specific antibodies; the ratio of each phospho-MAPK to total-MAPK was calculated to control for equal loading. (**G**) Protein bands were densitometrically quantified and analyzed using ImageJ software. Results are presented as means ± SD, expressed as a percentage relative to controls (* *p* < 0.05, ** *p* < 0.01, *** *p* < 0.001 vs. control group; # *p* < 0.05, ### *p* < 0.001 vs. the α-MSH-untreated control group).

**Figure 6 ijms-23-14922-f006:**
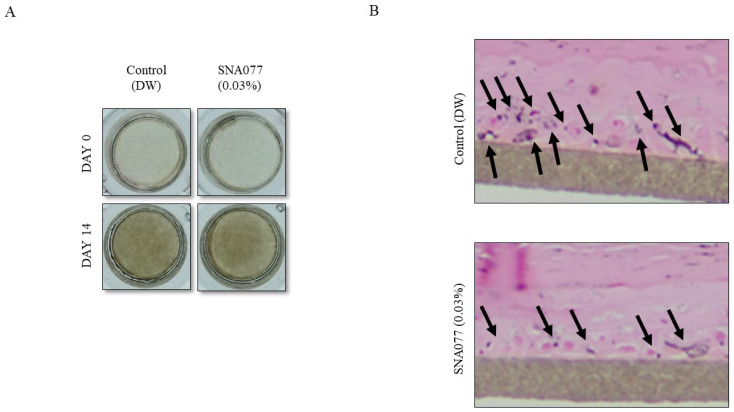
Inhibitory effect of SNA077 on the pigmented human skin model, MelanoDerm. (**A**,**B**) Skin whitening effects of SNA077 on a 3D pigmented epidermis skin model. MelanoDerm was treated with different concentrations of SNA077 for 14 days, and epidermal pigmentation was assessed by optical evaluation (**A**) and histological analysis using a Fontana Masson’s staining kit (**B**).

## Data Availability

The data that support the findings of this study are available from the corresponding author upon reasonable request.

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
