# Peer review of "SNA077, an Extract of Marine Streptomyces sp., Inhibits Melanogenesis by Downregulating Melanogenic Proteins via Inactivation of cAMP/PKA/CREB Signaling"

_ijms, 2022, doi:10.3390/ijms232314922_

Round 1
Reviewer 1 Report
The manuscript titled << SNA077, an Extract of Marine Streptomyces sp., Inhibits Mel- 2 anogenesis by Downregulating Melanogenic Proteins via Inac- 3 tivation of cAMP/PKA/CREB Signaling>> seems interesting. However, please consider these commnets.
- The chromatogram of the extract should be shown in the main text together with the identified compounds whether isolated or not. The level of identification should be considered and documented.
- The UV is not a tool for identification. You cannot identify the metabolites using UV only.
- Also, I would like to confirm what the active metabolites responsible for such an activity of the extract.
- From the medicinal point of view, can I obtain large amount of the extract in an easy and <<< reproducible >>> way if I would like to apply your finding as a medicine.
Thank you !
Author Response
Dear Reviewer
Thank you for kind review.
Now, we suggest the response to reviewer. And the corrected the sentences were highlighted with red color text in the revised manuscript.
It is our earnest hope that our efforts will meet your approval and be worthy of publication in IJMS. Your kind attention is appreciated in advance.
Sincerely yours,

Reviewer 2 Report
Comments to Authors
This paper presents evidence that Streptomycetes extract inhibits melanogenesis by interfering with MC!R signaling. There are several points that should be clarified.
1. Abstract: The first sentence makes very little sense. Skin pigmentation does not represent the most dangerous form of skin cancer; melanoma cells represent the most dangerous form. And since your paper has nothing to do with skin cancer, that statement needs to be deleted.
2. Introduction: Similarly, the first sentence stating that “To improve the hyper-pigmentary disorders such as melanoma….” Is incorrect. Research is never done to find depigmenting agents to treat melanoma. Looking for depigmenting agents is primarily done to find cosmetic solutions to hyperpigmentation. So delete reference to melanoma.
3. Results
a. 2.2 The authors state that 25ug/ml of the extract caused only “mild toxicity”. But from figure 1, while the toxicity was mild for B-16 mouse melanoma, it was very high for the human melanoma line, MNT. This needs to be clarified.
b. 2.2 For the experiments to look at the effect of the SNA077 extract on tyrosinase activity directly in a “test tube” assay, the authors only looked at mushroom tyrosinase. This enzyme is very different from the mammalian form. The correct experiment would have been to prepare an extract from untreated mouse or human melanoma cells and then test SNA077 for its ability to inhibit tyrosinase activity in this extract. The dopa oxidase assay could be easily used to test whether or not the SNA extract could directly inhibit mammalian tyrosinase. Typically, kojic acid is used as a positive control. The experiment cannot be done by treating cells with the SNA extract and then measuring tyrosinase in the subsequent extract. This does NOT determine whether SNA077 can directly inhibit the enzyme. This experiment needs to be done.
c. 2.2. In every experiment with B-16 melanoma cells the authors stimulated the cells with MSH in the presence or absence of SNA077. I saw no experiments where untreated B-16 cells were treated with the extract. In other words, what does the extract do to tyrosinase activity levels in non-stimulated mouse or human melanoma cells? It seems like this control should have been included in these studies.
d. 2.2 In Figure 1 panel F and panel G, why were different concentrations of the extract used for “cell viability” and “melanin content”. In F the concentrations were 6.25, 12.5, and 25 ug/ml whereas in panel G the concentrations were 2.5, 5, and 10ug/ml. No explanation was given.
e. 2.3 In panel A all three melanogenic proteins, tyrosinase, TRP-1 and TRP-2 were quantified, but in MNT cells, panel B, only tyrosinase and TRP-1 were examined. Why was TRP-2 mRNA not determined? In panel G all three melanogenic proteins were measured in MNT cells
f. 2.3 In figure 3, panel F the data suggests that SNA077 inhibits MITF protein expression but the actual blot is unconvincing. The band is distorted and this could lead to an artificially low density measurement.
g. 2.4 In the studies to measure cAMP levels the measurements were taken 15 minutes after adding MSH and plus or minus SNA077. The measurement of MC1R mRNA was taken 10 minutes after adding MSH/SNA077. The authors suggest that the low cAMP levels in cells treated with SNA077 might be due to a reduction in MC1R levels. This is very unlikely since the B16 cells have a steady state level of MC!R receptor and just inhibiting the mRNA for MC1R would not cause the reduction in cAMP seen at 15 minutes after MSH addition. It is most likely that the SNA077 is somehow inhibiting adenylate cyclase or some intracellular signal needed to activate adenylate cyclase. In vitro assays of adenylate cyclase activity in membrane preparations treated with SNA077 would have been interesting.
Author Response

(The authors gave the same response as above.)

Round 2
Reviewer 1 Report
I do recommend its acceptance.